Subject Areas:
biomedical engineering/neuroscience/computer modelling and simulation

Keywords:
tightrope balancing, intermittent control, simulation, ankle strategy, hip strategy

Author for correspondence:
Pietro Morasso
e-mail: pietro.morasso@iit.it

# Centre of pressure versus centre of mass stabilization strategies: the tightrope balancing case

## Pietro Morasso

Department of Robotics, Brain and Cognitive Sciences, Istituto Italiano di Tecnologia, Via Enrico Melen 83, 16152 Genoa, Italy

iD PM, 0000-0002-3837-8004

This study proposes a generalization of the ankle and hip postural strategies to be applied to the large class of skills that share the same basic challenge of defeating the destabilizing effect of gravity on the basis of the same neuromotor control organization, adapted and specialized to a variable number of degrees of freedom, different body parts, different muscles and different sensory feedback channels. In all the cases, we can identify two crucial elements (the CoP, centre of pressure and the CoM, centre of mass) and the central point of the paper is that most balancing skills can be framed in the CoP–CoM interplay and can be modelled as a combination/alternation of two basic stabilization strategies: the standard well-investigated *COPS* (or CoP stabilization strategy, the default option), where the CoM is the controlled variable and the CoP is the control variable, and the less investigated *COMS* (or CoM stabilization strategy), where CoP and CoM must exchange their role because the range of motion of the CoP is strongly constrained by environmental conditions. The paper focuses on the tightrope balancing skill where sway control in the sagittal plane is modelled in terms of the *COPS* while the more challenging sway in the coronal plane is modelled in terms of the *COMS*, with the support of a suitable balance pole. Both stabilization strategies are implemented as state-space intermittent, delayed feedback controllers, independent of each other. Extensive simulations support the degree of plausibility, generality and robustness of the proposed approach.

## 1. Introduction

Fighting against gravity is one of the main challenges that are faced by humans in a variety of daily life activities as well as sport performances, with a wide range of perceived difficulty [1]. For example, we may locate upright standing over a rigid surface on

one extreme of the range, while balancing and walking on a tightwire on the other extreme. For healthy adult subjects the skill necessary for managing the former task is an ability acquired early in life and performed in a fully automatic manner without any degree of attentional effort; by contrast, the latter task involves a demanding process of skill acquisition and learning. In spite of this, there is ground to believe that the two skills, as well as a variety of other skills that share the same basic challenge of defeating the destabilizing effect of gravity, are based on the same neuromotor control organization, adapted and specialized to variable number of degrees of freedom, different body parts, different muscles and different sensory feedback channels. In all the cases, we can identify two crucial elements: the centre of pressure (CoP) and the centre of mass (CoM). For any balancing task, the CoM position is always *above* the CoP and this identifies a potentially unstable *Inverted Pendulum* (IP), with an (unstable) equilibrium state characterized by the fact that CoM and CoP are aligned on the same vertical.

Although there are particular situations in which equilibrium emerges automatically in a purely passive way, in all the balancing paradigms of interest equilibrium is achieved in an active manner by selecting the appropriate groups of muscles and activating them with well-timed activation patterns. Consider, for example, a person sleeping in bed: he/she is not aware of the distribution of contact forces (i.e. the CoP position) and is unaware as well of the position of the CoM, but whatever unintentional movement is performed, the contact forces will be automatically redistributed in such a way to realign the CoP with the CoM without any motor planning/control. By contrast, in the case of quiet bipedal upright standing, the natural tendency to fall is counteracted mostly by the feedback activation of ankle muscles, supplementing the relevant but insufficient contribution of ankle muscle stiffness [2–4]. Moreover, the stiffness or impedance control paradigm [5–7], that requires active co-activation of antagonist muscles, contradicts the minimum intervention principle [8], which is a general organizing feature of many biological systems. In any case, the feedback activation of ankle muscles is driven by the sensory detection of ongoing sway and the biomechanical effect of this control action is to move the CoP inside the basis of support (BoS), i.e. the convex hull that encloses all points of contact between the feet or, more generally, the body, and the supporting surface. The *bounded* stability that characterizes postural sway in upright standing [9–13] requires that the peaks of the CoP profile overcome the corresponding peaks of the CoM profile with the hard constraint of remaining inside the BoS. We may call such active control paradigm *COPS* (CoP stabilization strategy) and clearly, its chance of success is a function of the size/shape of BoS: in upright standing, it decreases if the feet are close to each other or, even more, if one is positioned in front of the other (tandem position). Ultimately, the *COPS* paradigm is bound to fail as the width of the support base tends to vanish, thus forcing the brain to adopt a paradigm shift, as explained in the following.

In the *COPS* framework, the CoM is the controlled variable, stabilized by the modulation of the CoP, which plays the role of the control variable in the feedback loop. Such control strategy applies to standard upright standing but also to less familiar situations like handstanding [14–17] or inverted stick balancing on a fingertip [18–20]: in the former case, the ankle joint is substituted by the wrist joint, while, in the latter, by the contact point of the stick on the finger. However, in both cases, as in upright standing, the brain must learn to shift the CoP, back and forth and side to side, with the appropriate timing and amplitude, in order to keep the CoP–CoM virtual inverted pendulum up and avoid a fall. Moreover, in handstanding, the CoP is shifted indirectly by modulating the activation of wrist muscles, with a BoS related to the hand size, whereas in stick balancing the CoP is shifted directly by moving the hand and the BoS is related to the range of motion of the arm/body complex. By contrast, there are balancing tasks where the CoP is not the appropriate control variable because its physical/physiological range of motion is very limited or virtually null at least in one direction. One example is balancing the standing body on a tightwire or a very narrow bar: in both cases, the BoS is very narrow in one direction while it is sizable in the other. Thus, a tentative tightrope walker, after aligning his/her sagittal plane with the tightrope, will naturally manage to constrain the sway of the body in the sagittal plane by adopting the *COPS* as when standing on earth, but will soon discover that this strategy is totally ineffective in the coronal plane. In such situations, what the brain can do for avoiding the fall is to invert the roles of the CoP and CoM information in the stabilization process: instead of modulating the CoP position, for constraining the sway of the CoM, the brain can modulate the CoM position in order to keep it as close as possible to the vertical line centred on the virtually unmovable CoP. We may call this balancing strategy *COMS* (CoM stabilization strategy) which may be implemented in a variety of manners and variety of styles by recruiting and moving body parts or artificial tools.

There is no doubt that the experimental and theoretical literature related to *COPS* is much wider than the one related to *COMS*. The present study focuses on the skill of balancing on a tightwire, which

exemplifies many of the crucial control problems and is characterized by the fact that the two strategies must coexist and/or cooperate: COPS for controlling sway in the sagittal plane and COMS in the coronal plane. For this purpose, a computational model is developed by extending previous work based on the intermittent feedback control paradigm developed for the stabilization of standard upright standing and cart inverted pendulum [21–26]. Such studies were all formulated in the COPS framework whereas this paper introduces, for the first time, the formalization of the COMS model and its integration with the COPS in the same combined skill.

In particular, the state-space intermittent stabilization paradigm is exported from the COPS to the COMS domain. In both cases, the intermittency of the control action is motivated by the opportunity to exploit the *dynamic affordance* provided by the saddle-like instability of an inverted pendulum. With this type of instability it is possible to identify a stable and an unstable manifold in each of the two-phase planes of the oscillating IP (one for the sagittal sway and another for the coronal sway). Each phase plane can be divided into four regions: two fully unstable or unsafe regions and two meta-stable or safe regions. If the state vector of the uncontrolled IP enters one of the unsafe regions it will monotonically diverge from equilibrium until fall; in the other case, the IP would spontaneously approach equilibrium under the action of the stable manifold: this is the affordance provided by the saddle-type instability for a state-space intermittent feedback controller. In particular, such dynamic affordance suggests a natural switching rule for exploiting it: as long as the current estimate of the state vector remains inside a safe region, the controller may turn off any control action (off-phase), letting the pendulum evolve at its natural pace, whereas it should switch on the feedback control action as soon as the state vector enters one of the unsafe regions (on-phase). What is important is that, during the on-phase, the purpose of the control action is not to attract the state vector towards the nominal equilibrium state but to the stable manifold. The bounded stability that can be achieved with this intermittent feedback approach is quite robust because it can work also with delayed information of the state vector, producing a limit cycle as an alternation of segments of hyperbolic orbits (off-phases) and spiral orbits (on-phases). As reported in the previous studies of the COPS paradigm [21–26], the control action during the on-phases is a linear combination of the current (delayed) estimate of the state vector and this control action drives the actuating force/torque that determines a motion of the CoP. With the newly investigated COMS paradigm, based on the assumption of a virtually unmovable CoP, the intermittent control action is aimed at shifting laterally the balance pole for counteracting the body tilt.

Considering that the scientific literature on tightrope balancing and related skills is rather scarce [27–29], this study intends to provide a preliminary first-order framework for designing future more specific and precise experimental and theoretical studies. The model includes several simplifications in order to keep it tractable to a simulation study: (i) the body is schematized as an inverted pendulum, swaying in two planes but ignoring multi-joint coordination; (ii) the motor control of the two arms holding the balance pole is reduced to a medio-lateral shift of the pole, perpendicular to the body axis, without rotation; (iii) the balance pole is assumed to be rigid and straight; (iv) the pole is very long, in agreement with the length chosen by professional long-distance tightrope walkers; (v) reaction forces are not taken into account; and (vi) the pole is only shifted medio-laterally, not rotated around an antero-posterior axis. In the following sections, the rationale of such assumptions will be explained, but I also wish to state, right away, that I think that in spite of such simplifications the model is likely to capture the key issues, in order to answer the following crucial question: to what extent can the hybrid dual intermittent controller, based on the CoP and the CoM strategies, succeed to assure bi-axial stability in a robust and independent way, without any specific optimal coordination/synchronization of the two control modules?

## 2. Methods

The hypothetical tightrope walker is represented as an inverted pendulum, swaying on two planes (sagittal and coronal) thus identifying the two degrees of freedom of the model

$$q = \begin{bmatrix} q_{ap} \\ q_{ml} \end{bmatrix}. \tag{2.1}$$

The feet are assumed to be in a tandem position and aligned with an infinite horizontal rigid wire that supports them. A reference frame is considered, with the $y$-axis along the wire, the $z$-axis vertical and the $x$-axis characterizing the medio-lateral direction of the body. The origin of the frame is a point on the wire

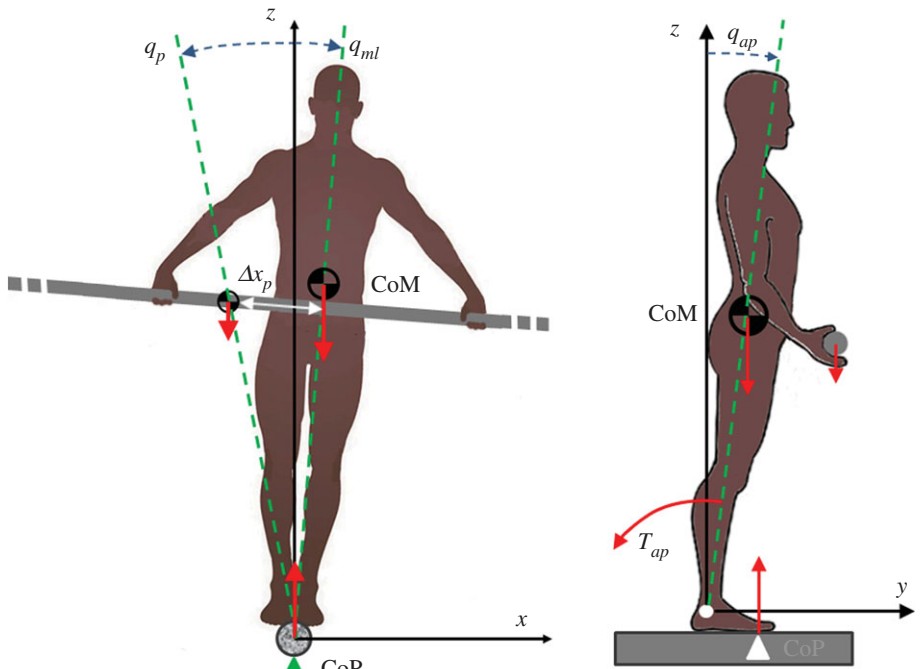

**Figure 1.** The body of the tightrope walker is approximated as an inverted pendulum swaying in two planes: sagittal plane ($q_{ap}$) and coronal plane ($q_{ml}$). The CoP can shift back and forth along the y-axis but is unmovable along the x-axis. The walker can move sideways the long balancing pole, with respect to the body axis ($\Delta x_p$), thus implementing the *COMS* in the coronal plane. The *COPS* is implemented in the sagittal plane by activating the ankle torque ($T_{ap}$) that determines the anterior–posterior shift of the CoP.

related to the ankle position of the backward foot. The CoP can move forward–backward along the wire but is unmovable medio-laterally. The walker holds a balancing pole, at the height of the body CoM, that is actively moved sideways ($\Delta x_p$) for implementing the *COMS* in the coronal plane; the *COPS* is implemented in the sagittal plane by activating the ankle torque ($T_{ap}$) that determines the anterior–posterior shift of the CoP (figure 1).

The balance pole is characterized by a length $l_p$ and a mass $m_p$ and, for simplicity, it is assumed that it is rigid and straight. At equilibrium, the position of the pole-CoM along the x-axis coincides with the body-CoM; otherwise, the pole is shifted sideways, according to the displacement $\Delta x_p$, in order to compensate an incipient lateral fall of the body. The global inertia tensor (body + balance pole) is changed as a function of the medio-lateral shifts of the pole carried out by the active stabilization actions, but since the range of such displacement is very small (of the order of cm), the tensor will be assumed to be constant. The moment of inertia of the pole is neglected for rotations around its longitudinal axis and is computed by the following formula for any axis through the CoM of the pole, perpendicular to it:

$$I_p = \frac{1}{12} m_p\, l_p^2. \tag{2.2}$$

For the body itself, the available numerical estimates of the human body inertia tensor [30,31] are usually expressed in terms of a reference frame centred in the body CoM. For computing the inertia tensor related to a frame centred on the feet, at contact with the supporting wire, it is possible to use the parallel axis theorem, ending up with the following equation for the global inertia tensor:

$$J = \begin{bmatrix} J_{xx} & J_{xy} \\ J_{yx} & J_{yy} \end{bmatrix} = J_{\text{body}} + J_{\text{pole}} \quad \begin{cases} J_{\text{body}} = \begin{bmatrix} I_{xx} & I_{xy} \\ I_{yx} & I_{yy} \end{bmatrix} + h_{\text{com}}^2 \begin{bmatrix} m_b & 0 \\ 0 & m_b \end{bmatrix} \\ J_{\text{pole}} = \begin{bmatrix} 0 & 0 \\ 0 & I_p \end{bmatrix} + h_{\text{com}}^2 \begin{bmatrix} 0 & 0 \\ 0 & m_p \end{bmatrix} \end{cases}. \tag{2.3}$$

Here $J_{ij}$ are the elements of the global, wire-centred tensor, $I_{ij}$ are the elements of the CoM-centred tensor of the body in the standard posture and $h_{\text{com}}$ is the distance of the CoM from the feet. The dynamics of the tightrope walker is then characterized by the following equation, where the effect of the Coriolis and centrifugal components is ignored because the sway movements are slow and of

small amplitude:

$$J \ddot{q} = T_g(q) - T_s(q, \dot{q}) - T_{int}(q_\delta, \dot{q}_\delta) + wn. \tag{2.4}$$

Thus the time course of the sway motion $q = q(t)$ is determined by three torque vectors, in addition to a white noise disturbance: the destabilizing gravitational torque $T_g$, counteracted by a *passive* stiffness-related component $T_s$ and an *active* intermittent component $T_{int}$; $T_g$ and $T_s$ depend on the current value of the state vector, whereas $T_{int}$ is a function of the delayed estimate of the state vector. For the computation of $T_g$ it is assumed, for simplicity, that the balance pole is kept at the height of the body-CoM and its axis, parallel to the $x$-axis, passes through the body-CoM; moreover, the equation is linearized around the equilibrium posture

$$T_g = \begin{bmatrix} G_{ap} \\ G_{ml} \end{bmatrix} = \begin{bmatrix} (m_b + m_p)\, g\, h_{com}\, q_{ap} \\ m_b\, g\, h_{com}\, q_{ml} \end{bmatrix}. \tag{2.5}$$

$T_s$ is related to the mechanical properties of ankle muscles (stiffness $K_a$ and viscosity $B_a$) and clearly is limited to the sway component in the sagittal plane because ankle muscles cannot play any role for the stabilization in the coronal plane

$$T_s = \begin{bmatrix} K_a q_{ap} + B_a \dot{q}_{ap} \\ 0 \end{bmatrix}. \tag{2.6}$$

In agreement with available estimates of ankle stiffness [3,4] $K_a$ is smaller than the rate of growth of the gravity toppling torque in the sagittal plane

$$K_a < (m_b + m_p)\, g\, h_{com}. \tag{2.7}$$

$T_{int}$ expresses the active control mechanism that is based on a state-based delayed-intermittent-feedback control action, different for the two degrees of freedom: the sway component in the sagittal plane is stabilized according to the same *COPS* paradigm that modulates ankle muscles in quiet upright standing, whereas the more critical sway in the coronal plane relies on the newly investigated *COMS* paradigm, namely the skilled handling of the balancing pole

$$T_{int} = \begin{bmatrix} T_{ap} \ (COPS \ \text{based}) \\ T_{ml} \ (COMS \ \text{based}) \end{bmatrix}. \tag{2.8}$$

*COPS*: in this case, the control action is direct, i.e. it involves the intermittent activation of ankle muscles in order to produce an ankle torque $T_{ap}$, as a function of the current (but delayed) estimate of the state vector $[q_{ap}(t - \delta),\ \dot{q}_{ap}(t - \delta)]$ in the phase plane of the sagittal sway. The switching rule from the on-phase to the off-phase and back is formulated in the state-space of the oscillating body in the sagittal plane:

<u>**On-phase**</u>
  **Activation condition**: $q_{ap}(t - \delta)\,[\dot{q}_{ap}(t - \delta) + \alpha\, q_{ap}(t - \delta)] \geq 0$
  **Control Action**: $T_{ap}(t) = P_{ap}\, q_{ap}(t - \delta) + D_{ap}\, \dot{q}_{ap}(t - \delta)$

<u>**Off-phase**</u>
  **Dis-activation condition**: $q_{ap}(t - \delta)\,[\dot{q}_{ap}(t - \delta) + \alpha\, q_{ap}(t - \delta)] < 0$
  **Control Action**: $T_{ap}(t) = 0$

$$(2.9)$$

*COMS*: in this case, the control action is indirect, i.e. it activates the arm muscles in order to induce a lateral shift $\Delta x_p$ of the pole-CoM that generates a gravity torque $T_{ml}$ opposed to the destabilizing torque arising from the tilted body-CoM. In the specific implementation of this mechanism, reported here, the stabilizing torque is obtained in two steps: (i) a control variable $\gamma$ is computed, as a function of the current (but delayed) estimate of disequilibrium $[q_{ml}(t - \delta),\ \dot{q}_{ml}(t - \delta)]$ in the coronal plane; and (ii) the time profile of the control variable $\gamma(t)$, which is discontinuous because the active control is intermittent, is smoothed out by means of a simple low-pass filter (LPF), thus generating the smoothed commanded displacement $\Delta x_p$ of the balance pole that immediately induces the stabilizing torque $T_{ml}$. The switching from the on- to the off-phase and back is formulated in the state-space of the oscillating body in the coronal plane:

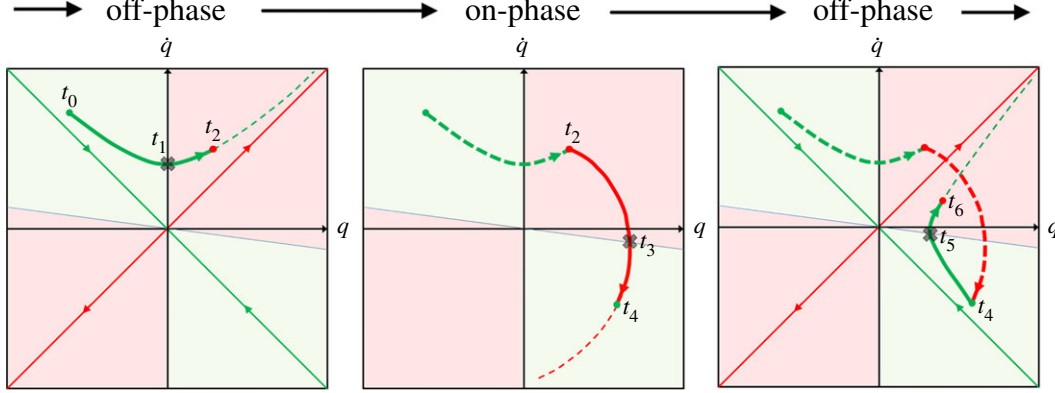

**Figure 2.** Illustration in the phase plane of the intermittent stabilization strategy for both controllers: $q$ stands for $q_{ap}$ in the case of COPS and $q_{ml}$ in the case of COMS. The three panels represent the alternation from an off-phase (initiating at $t = t_0$) to an on-phase and then to an off-phase (terminating at $t = t_6$). The reddish filled areas correspond to the *unsafe regions* and the greenish filled areas to the *safe regions*. The unsafe regions include the first and third quadrants of the phase plane plus a small triangular slice; the safe regions complement the phase plane area. The parameter $\alpha$ of equations (2.9) and (2.10) is the slope of the line that separates the safe from the unsafe regions. In the off-phases, the origin is a saddle point and the dynamics is characterized by the stable manifold (green line) and the unstable manifold (red line). Starting in the off-phase at $t = t_0$ the ideal evolution of the state vector, in the absence of noise, is the thick green trajectory (a hyperbolic orbit); at $t = t_1$ the state vector enters the unsafe area but due to the feedback delay the activation condition of the on-phase occurs only at $t = t_2 = t_1 + \delta$. The activation of feedback control (initiation of the on-phase) avoids that the state is pushed to a fall by the unstable manifold and the following orbit will be spiral-like, crossing the boundary between unsafe and safe region at $t = t_3$; the transition to the following off-phase will only occur later on at $t = t_4 = t_3 + \delta$ and the corresponding segment of hyperbolic orbit will terminate at $t = t_6 = t_5 + \delta$. Bounded stability can be achieved even if the spiralling orbits in the on-phases are globally unstable: it is sufficient that, on average, such orbits terminate 'close enough' to the stable manifold.

---

**On-phase**
 **Activation condition**: $q_{ml}(t - \delta) \left[\dot{q}_{ml}(t - \delta) + \alpha\, q_{ml}(t - \delta)\right] \geq 0$
 **Control variable**: $\gamma(t) = -P_{ml}\, q_{ml}(t - \delta) - D_{ml}\, \dot{q}_{ml}(t - \delta)$

**Off-phase**
 **Dis-activation condition**: $q_{ml}(t - \delta) \left[\dot{q}_{ml}(t - \delta) + \alpha\, q_{ml}(t - \delta)\right] < 0$
 **Control variable**: $\gamma(t) = 0$

 **Control Action**
 $\Delta x_p = LPF(\gamma) \rightarrow q_p = \Delta x_p / h_{com} + q_{ml} \rightarrow T_{ml}(t) = m_p g\, h_{com}\, q_p$

$$(2.10)$$

More specifically, the control variable $\gamma(t)$ consists of displacing the pole CoM in the opposite direction of the body CoM disequilibrium ($\gamma = -(Pq + D\dot{q})$) during the on-phase, whereas the two CoMs are kept aligned in the off-phase ($\gamma = 0$). Since $\gamma$ is clearly a discontinuous variable it will be smoothed out by the arm control system. For simplicity, this smoothing is implemented in the simulation package by a simple second-order LPF, which is characterized by the following transfer function: $F(s) = 1/((s/\omega_f)^2 + 2\xi(s/\omega_f) + 1)$, with a cut-off frequency of 2 Hz and a critical damping factor ($\xi = 0.7$). The control action, namely the generation of the torque $T_{ml}$ supposed to compensate the destabilizing effect of gravity, is then the biomechanical consequence of this controlled shift: note that, as shown in figure 1, $q_p$ is the angle between the line from the rope to the pole CoM and the vertical.

As emphasized above, the stabilizing torque in the COPS case is the direct effect of ankle muscle activation whereas in the case of COMS it is the biomechanical consequence of a tool/body shift. In both cases, however, the rationale of the intermittent activation paradigm is to take advantage of the affordance provided by the saddle-type instability that characterizes the dynamics of an inverted pendulum, namely the existence of stable and unstable manifolds in the phase plane. As previously remarked, this dynamics is characterized by the fact that the phase-plane can be divided into *safe* and *unsafe* regions (greenish and reddish areas, respectively, in figure 2). The activation/dis-activation

**Table 1.** Anthropometric and control parameters.

| model parameters | parameter type | parameter used in the simulations | range of control parameters for stability |
|---|---|---|---|
| $m_b$ (kg) | body mass | 78 | |
| $h_{com}$ (m) | height of the body CoM | 0.997 | |
| $m_p$ (kg) | mass of the balance pole | 13 | |
| $l_p$ (m) | length of the balance pole | 12 | |
| $I_{xx}$ (kg m$^2$) | body inertia tensor | 13.496 | |
| $I_{xy} = I_{yx}$ (kg m$^2$) | body inertia tensor | 0.0344 | |
| $I_{yy}$ (kg m$^2$) | body inertia tensor | 14.690 | |
| $\delta$ (s) | feedback delay | 0.2 | |
| $\alpha$ (s$^{-1}$) | slope of the line separating safe and unsafe regions in the state plane | 0.4 | |
| $\omega_f$ (Hz) | cut-off frequency of the smoothing filter | 2 | |
| $K_a$ (N m rad$^{-1}$) | ankle stiffness | 495.87 | |
| $B_a$ (N m s rad$^{-1}$) | ankle viscosity | 20 | |
| $P_{ap}$ (N m rad$^{-1}$) | proportional control parameter | 578.52 | 445−1335 |
| $D_{ap}$ (N m s rad$^{-1}$) | derivative control parameter | 20 | <500 |
| $P_{ml}$ (m rad$^{-1}$) | proportional control parameter | 9 | 7−10 |
| $D_{ml}$ (m s rad$^{-1}$) | derivative control parameter | 8 | <10 |

conditions of equations (2.9) and (2.10) correspond to the fact that the trajectory of the delayed state vector enters one region coming from the other (figure 2). The parameter $\alpha$ modifies the border between the safe and unsafe regions of the state plane and was introduced in a previous study [23] for taking into account that the feedback information on the state vector is made available to the controller with a significant delay. The theoretical and numerical analysis carried out in the previously quoted paper demonstrated indeed that the robustness of the intermittent feedback action is optimized by choosing $\alpha = 0.4$ (for the feedback delay $\delta = 200$ ms), in the sense that it minimizes the sensitivity to variations of the control parameters. Moreover, it was also demonstrated that this type of discontinuous feedback control can achieve bounded stability even if the alternated regimes (with and without feedback activation) are unstable if taken in isolation. However, while the efficacy of the intermittent control paradigm has been investigated at large for the CoP strategy this is not the case for the CoM strategy which is investigated here in a simplified manner, by focusing only on the *static* stabilizing effect of the displaced pole-CoM while ignoring the dynamic effect due to the action–reaction principle. In order to reduce such dynamic effect the intermittent control variable is low-pass filtered before producing the stabilizing gesture that, in any case, generates a quite slow sway oscillation due to the large inertia provided by the balance pole.

The anthropometric parameters used by the model were selected with reference to standard evaluations [30,31] and are listed in table 1. For the viscous-elastic parameters of the ankle joint ($K_a$ and $B_a$) the used values were derived from the previous study [23]. The control loop delay $\delta$, including the sensory and motor components, was set equal to 200 ms.

For the balance pole the adopted numeric values ($l_p = 12$ m, $m_p = 13$ kg) were derived from the choices of famous professional tightrope walkers, like Nick Wallenda or Philippe Petit, in very challenging enterprises. Let us explain the rationale of this choice. Considering the oscillations of the body around the $y$-axis, for the anthropometric parameters of table 1 and the long balance pole mentioned above, the moments of inertia of the body and the pole have the following values:

$$\begin{cases} J_{\text{body}} = 92.22 \\ J_{\text{pole}} = 168.92 \end{cases} \text{kg m}^2. \tag{2.11}$$

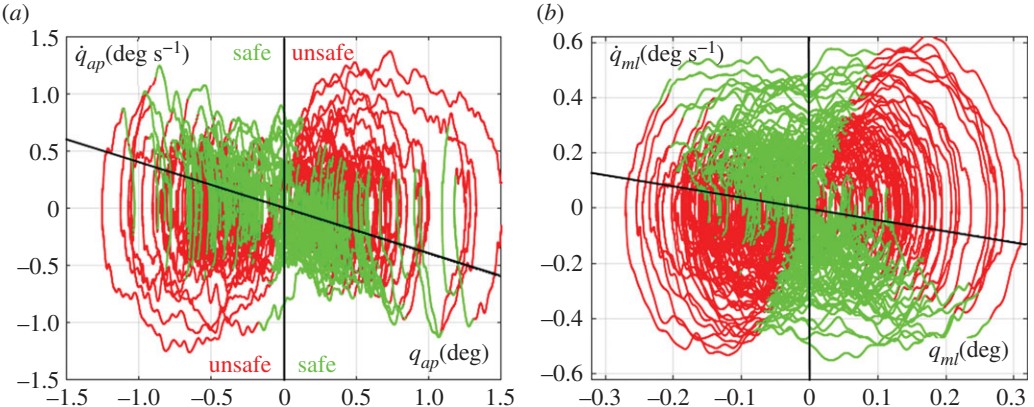

**Figure 3.** Simulated sway movements of the tightrope balancing paradigm. (*a*) Phase portrait of sway movements in the sagittal plane, controlled with the CoP strategy. (*b*) Phase portrait of sway movements in the coronal plane, controlled with the CoM strategy. Both graphs cover a 4 min time period. The phase planes are divided in two *safe* and two *unsafe* regions, delimited by the vertical line and the line with the slope −α. The green-coloured orbit segments correspond to the off-phases and the red-coloured segments to the on-phases, respectively. Please note the different scale factors in the two graphs: the amplitude of oscillation in the *ap* direction is about three times the amplitude in the *ml* direction.

This means that the overall moment of inertia around the *y*-axis is almost three times larger than the moment of inertia of the naked body, with a significant increase of the falling time constant

$$\begin{cases} \tau_{\text{body}} = \sqrt{\dfrac{J_{\text{body}}}{m_b\, g\, h_{\text{com}}}} = 0.3477 \text{ s} \\ \tau_{\text{total}} = \sqrt{\dfrac{J_{\text{body}} + J_{\text{pole}}}{(m_b + m_p)g\, h_{\text{com}}}} = 0.5417 \text{ s} \end{cases}. \tag{2.12}$$

Consider now a half-size balance pole (half-length and half-weight): the moment of inertia of the pole is reduced by almost 90% and the total time constant becomes 0.38 s, i.e. quite close to intrinsic time constant of the naked body. Clearly this justifies the empiric choice of the parameters of the balancing pole performed by the above-mentioned professional tightrope walkers.

# 3. Results

The computational model described in the previous section was simulated by means of Matlab® (MathWorks), using the forward Euler method for integrating the second order differential equation (equation (2.4)) with a time step of 1 ms and with the anthropometric and control parameters listed in table 1. The Matlab simulation script can be downloaded. Since the model includes a noise term (the white noise disturbance *wn* applied to equation (2.4)) the model is a stochastic differential equation (SDE) and thus its solution, i.e. *q*(*t*), is a stochastic process. Instead of using the sophisticated techniques of integration of SDEs, which are difficult to apply in this case due to the strong nonlinearity introduced by the intermittent controllers, the solution adopted in the simulation experiments is based on the rationale of Monte Carlo experiments (a class of computational methods that rely on repeated random sampling to obtain numerical results): this allows to estimate relevant statistic indicators of the underlying stochastic process by repeating the simulation with independent instantiations of the noise source.

The first test was about stability and robustness of the hybrid stabilization strategy: a specific model was considered stable if during three independent simulation runs of 240 s each there was no fall. Overall stability depends on the numerical values of the control parameters that bound the body sway in the sagittal plane by means of the CoP strategy ($P_{ap}$, $D_{ap}$) and the parameters of the more critical CoM strategy ($P_{ml}$, $D_{ml}$) that constrain the oscillations in the coronal plane. The simulations highlighted indeed the robustness of the proposed control because it was rather easy to empirically identify appropriate parameter values as well as rather ample admissible ranges for each of them: the values used in the simulations illustrated in the following figures are listed in table 1.

Figure 3 shows the phase portraits of the oscillations of the body inverted pendulum in the two planes, sagittal and coronal, respectively. The former phase portrait is quite similar to what has been

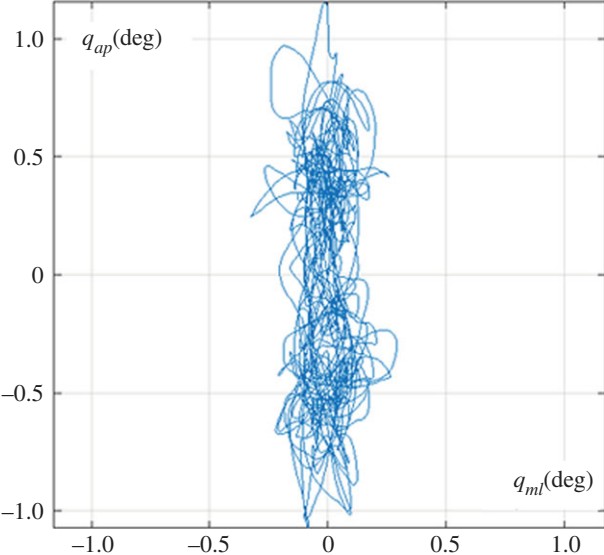

**Figure 4.** Simulated sway movements of the tightrope balancing paradigm: statokinesigram of the body oscillations, stabilized by the CoP strategy in the sagittal plane and the CoM strategy in the coronal plane.

found in the case of standard quiet upright standing in spite of the fact that the feet are not located side by side on the supporting element but one in front of the other, typical of the so-called tandem posture: this posture is known to worsen the standing stability [29] but only in the coronal plane. For the other phase portrait, I am not aware of specific quantitative studies to compare with the present result; in any case, the similarity of the two phase portraits, although with a different scale factor, is a consequence of the similar overall organization in spite of the strong difference in biomechanical and neuromotor control terms. The figure also illustrates the alternation between off- and on-phases by colouring differently the corresponding sway segments.

After having demonstrated the stability and robustness of the proposed hybrid stabilization strategy, it was worth evaluating the range of angular oscillation of the body in the two planes (evaluated as twice the standard deviation of the angular plots over a 240 s window, averaged over three simulation runs). Considering that the stabilization in the coronal plane is clearly the most critical part of the skill, as expected it was found that the range of angular sway in the sagittal plane is about three times larger than in the coronal plane: $R_{ap} = \pm 0.92$ deg versus $R_{ml} = \pm 0.31$ deg. The former indicator corresponds to a range of motion of the CoP ($R_{cop} = \pm 16$ mm) in the antero-posterior direction which is compatible with typical values recorded in upright standing. Figure 4 clarifies the combined sway patterns in the two planes, namely a kind of statokinesigram of the simulated tightrope walker: the two-dimensional plot of the body oscillation ($q_{ml}$ vs $q_{ap}$). It is worth mentioning that the statokinesigram in the standard upright standing situation, where the oscillations in both planes are controlled according to the CoP strategy, is close to isotropic and typically much less elongated. The CoP strategy can tolerate a rather large range of sway angles provided that the projection of the CoM remains inside the BoS. For the CoM strategy the BoS is virtually null and thus the risk of fall is much greater, forcing the tightrope walker to minimize the value of $R_{ml}$.

The intermittent control variable for stabilizing the sway in the sagittal plane is the active ankle torque $T_{ap}$ which operates according to the CoP strategy, supplemented by the stabilizing effect of the ankle stiffness provided by $T_s$: the standard deviation of the sum of the two stabilizing torques is $\sigma_{T_s + T_{ap}} = 7.47$ N m and it is obviously greater than standard deviation of the gravity destabilizing torque $\sigma_{G_{ap}} = 6.88$ N m. Such numerical predominance of the stabilizing torques with respect to the destabilizing effect of gravity is clarified also by figure 5a which shows a representative 30 s plot of the oscillations of the CoP and CoM in the antero-posterior direction ($y_{CoP}$ and $y_{CoM}$): the former variable is proportional to the combined stabilizing torque ($T_s + T_{ap}$) and the latter variable to the destabilizing torque $G_{ap}$. It appears that the two actions are in phase and the former one systematically overcomes the latter, thus producing a regime of bounded stability. Such dynamics fully agrees with what is known in standard posturographic analysis about the relationship between CoM and CoP plots, expressing the essence of the CoP strategy, where the CoP is the neuromotor control variable and the CoM is the biomechanical controlled variable.

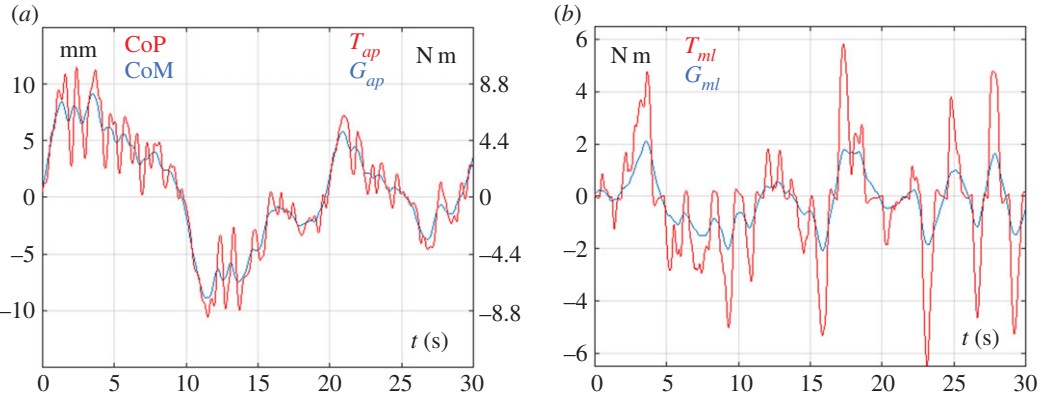

**Figure 5.** Simulated sway movements of the tightrope balancing paradigm: active intermittent control paradigms in the two planes. (a) Postural oscillations in the sagittal plane under the CoP control strategy: CoP is the control variable and CoM is the controlled variable; $T_{ap}$ and $G_{ap}$ are the corresponding torques. (b) Postural oscillations in the coronal plane under the CoM control strategy: $T_{ml}$ is the control torque compensating the destabilizing effect of the gravity torque $G_{ml}$.

For the sway in the coronal plane, stabilized according to the CoM strategy, the control variable is the lateral displacement of the balance pole $\Delta x_p$, directly and physically responsible of the control action $T_{ml}$ intended to counterbalance the gravity destabilizing torque $G_{ml}$. In a typical simulation, the standard deviation of $\Delta x_p$ is $\sigma_{\Delta x_p} = 2.61$ cm, which corresponds to a standard deviation of the control torque $\sigma_{T_{ml}} = 3.72$ N m, greater than the standard deviation of the gravity torque $\sigma_{G_{ml}} = 1.54$ N m. Figure 5b shows a representative 30 s plot of the two counteractive actions. Also in this case, the two actions are in phase, with the former one systematically overcoming the other in such a way to prevent the fall and push back the pendulum towards equilibrium. The simulations show that in the CoM strategy the amplitude of the control torque is about twice the gravity torque, whereas in the previously mentioned CoP strategy the supremacy of control versus gravity is much smaller (of the order of 10%). One reason is possibly the difference of the overall moments of inertia in the two cases: 91 kg m$^2$ versus 261 kg m$^2$.

The two angular oscillations ($q_{ap}$ and $q_{ml}$) are very weakly correlated (correlation coefficient (corr-coeff) of the order of 0.1), supporting the hypothesis that the two stabilization strategies can effectively function in an independent manner, in spite of the weak coupling determined by the inertia matrix and the potential entrainment of the two oscillatory patterns related to the intermittency of the control actions. By contrast, the angular oscillation in the sagittal plane $q_{ap}$ is strongly correlated to the corresponding control variable $T_{ap}$ (corr-coeff = 0.887) and the angular oscillation in the coronal plane $q_{ml}$ is strongly anti-correlated with respect to the control variable $\Delta x_p$ (corr-coeff = −0.957).

The intermittent control action in the sagittal plane (CoP strategy) is activated 65% of the time with a rate of 5.4 pulses s$^{-1}$ and an average duration of each pulse of 118 ms; in the coronal plane (CoM strategy) the control action is on 55% of the time with a rate of 4.2 pulses s$^{-1}$ and an average pulse duration of 130 ms.

The oscillatory regimes in the two planes are characterized by different spectral features (figure 6). The oscillations in the coronal plane are characterized by a clear resonant peak between 0.3 and 0.4 Hz, namely a behaviour which is similar to the stabilization of a cart inverted pendulum [18–20]: in spite of the obvious differences between the two cases, they share the fact that the oscillating body has the CoP fixed to the lower end of the pendulum, inhibiting any possible contribution of the CoP to its stabilization. By contrast, the sway in the sagittal plane is characterized by a typical $f^{-\beta}$ power spectral density (PSD) well known in the standard posturographic literature, where it is often modelled as a stochastic motion of the CoP acting on the foot soles during stance [32,33]. In particular, Collins & De Luca [32] showed that the spectral features of the temporal patterns of postural sway for healthy adults derive from the interplay between a short-term/high-frequency regime (about 0.7–10 Hz), where increments of the process behave as those of a positively correlated random walk, with a long-term/low-frequency regime (about 0.01–0.7 Hz), where increments of the CoP motion behave as those of a random walk with slightly negative correlation.

The robustness of the hybrid intermittent control strategy (*COPS* in one axis and *COMS* in the other) is well clarified by the fact that the bounded stability is preserved with a large range of variation of the four control parameters, as shown in table 1. In particular, it appears that the proportional parameters of

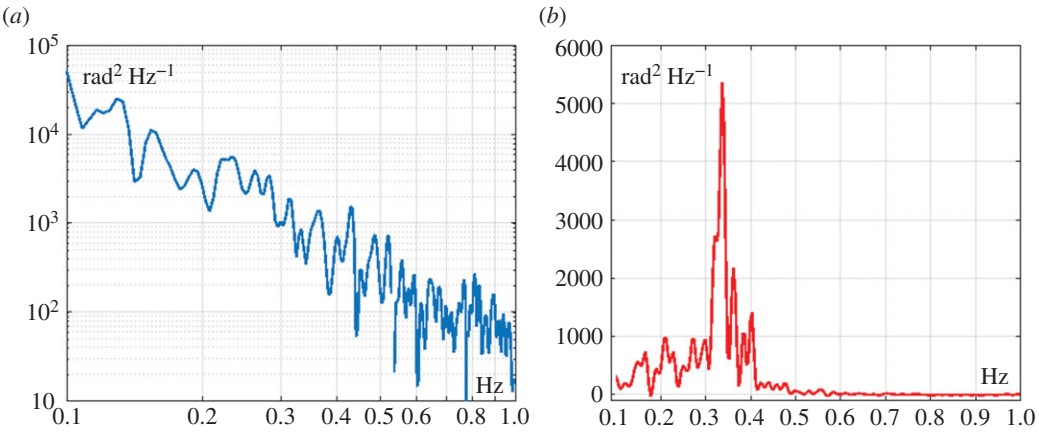

**Figure 6.** Simulated sway movements of the tightrope balancing paradigm. (*a*) Power spectral density of the postural oscillations in the sagittal plane ($q_{ap}$). (*b*) Power spectral density of the postural oscillations in the coronal plane ($q_{ml}$).

both control strategies ($P_{ap}$ for *COPS* and $P_{ml}$ for *COMS*) are more critical for stability than the derivative parameters ($D_{ap}$ and $D_{ml}$) which admit a very large range of values. Moreover, such range is greater for the CoP strategy, which admits a 50% variation around the mid-range value, than for the CoM strategy, where the percentage is less than 20%: this obviously agrees with the intuitive perception that in tightwire balancing the risk of fall is basically in the coronal plane. More specifically, we may consider that in the *COPS* case the gravity-driven risk of fall is counteracted by two effects: (i) ankle stiffness, which provides a continuous, undelayed physical feedback, and (ii) intermittent feedback control, based on delayed sensory information. By contrast, for *COMS* only intermittent delayed feedback is available, narrowing the range between the too small and too big feedback gain values capable to constrain oscillations to a bounded regime.

As previously mentioned, the physical parameters (length and weight) of the balancing pole used in the simulations are about a standard adopted by professional walkers for long, demanding walks. One may wonder to what extent such long poles are really necessary. In order to answer this question simulations were performed with shorter and proportionally lighter poles. It was found that with a pole length reduced to 60% of its original value it was impossible to avoid a quick fall, and for a 70% reduction, stability could be achieved but only for limited stretches of time (30–60 s). For a more limited pole reduction, stability over long stretches of time could be achieved but with a significant increase of the range of medio-lateral sway: as already stated, this range is $R_{ml} = \pm0.31$ deg for the regular balance pole and it is increased about 30% when the pole length is reduced by 10% and is further increased to more than 80% for a pole length reduction of 20%. Clearly, the results of such simulations provide a sound justification to the empirical choice of professional long-distance, high-wire walkers.

## 4. Discussion

The central point of the paper is that most balancing skills against gravity, whatever the number of involved body segments and degrees of freedom, can be framed in the CoP–CoM interplay and can be modelled as a combination/alternation of two basic intermittent stabilization strategies: the standard well-investigated *COPS* (the default option), where the CoM is the controlled variable and the CoP is the control variable, and the less-investigated *COMS*, where CoP and CoM must exchange their role because the range of motion of the CoP is strongly constrained by environmental conditions. The brain has direct/indirect access to both variables, through a number of noisy sensory channels, with the aid of complex multi-sensory data fusion processes that involve the central nervous system (CNS). In both cases, body sway is a random process characterized by bounded stability, rather than an asymptotic stability aiming at an ideal equilibrium point where CoM and CoP are aligned on the same vertical: the difference is that, in the *COPS* case, the CoP oscillation constrains the CoM orbit in the phase plane, keeping it inside; by contrast, in the *COMS* case, whole-body gestures (including hip, trunk and arms) aim to induce CoM oscillations that keep the corresponding orbit as close as possible to the constrained BoS.

The distinction between the two strategies above may be considered as a generalization of the well-known distinction between ankle and hip strategies for the control of sway movements in the sagittal

plane [34,35]. The common view is that the ankle strategy stabilizes the whole body oscillations as a single-segment inverted pendulum by producing a torque at the ankle that overcomes the gravity destabilizing torque, thus helping the 'inverted pendulum' to recover equilibrium: this dynamics is mirrored by the fact that the CoP position (proportional to the ankle torque) overcomes the position of the CoM projection (proportional to the gravity torque). The hip strategy, by contrast, activates the body as a double-segment inverted pendulum with counter-phase motion of ankle and hip. It is usually taken for granted that the single-segment, ankle strategy applies to healthy young subjects standing quietly on a rigid surface, whereas the hip strategy emerges in the case of unexpected postural perturbations, in elderly people or when standing on a soft base. Hip engagement for maintaining balance is an example of using the COMS strategy when environmental or physiological conditions reduce significantly the feasible range of motion of the CoP. However, this does not imply that, in general, engaging hip motion/torques mostly changes the CoM: hip torques could indeed change CoP through interjoint coupling if the BoS is sufficiently large. In other words, the clear-cut distinction between COPS and COMS is that in the former case the oscillation of the CoP is larger and anticipates the oscillation of the CoM, whereas in the latter case it is the other way around and it achieves stability by engaging the hip as well as other body parts.

However, we should also take into account the fact that the distinction above between a single inverted pendulum model for explaining the ankle strategy and a double inverted pendulum model for characterizing the hip strategy was somehow contradicted by several studies clearly showing that even in the case of quiet upright standing on a rigid surface hip joint rotations are not negligible in comparison with ankle rotations, particularly as regards angular velocity and acceleration [36–38]. This suggested that also for the ankle strategy the single inverted pendulum (SIP) model should be substituted by a multi-link model, at least a double inverted pendulum (DIP), involving the coordinated control of ankle and hip joints, capable to explain the characteristic coordinated kinematics of the two joints: a strong anti-correlation of the acceleration profiles accompanied by a weaker anti-correlation of the speed and a mild positive correlation of angular rotations. In particular, it was proposed [39] that such ankle–hip coordination patterns might be explicitly programmed by the CNS in order to minimize the amplitude of the CoM angular acceleration. An alternative solution [26] substituted the SIP with a virtual inverted pendulum (VIP) that links the ankle joint to the CoM, proposing a hybrid control of the VIP/DIP model: intermittent active control of the ankle (via the VIP part of the model) and passive stiffness control of the hip (via the DIP part). The simulations demonstrated that the proposed model is compatible with the complex inter-joint coordination patterns summarized above, without any explicit high-level coordination mechanisms. This is the main motivation for supposing that although the SIP model, adopted for simplicity in the present study, is indeed a simplification, it may be also considered a first-order approximation that enables us to clarify the role of the two fundamental stabilization strategies (COPS and COMS) and their coexistence in a task like tightrope walking.

As regards the use of the balancing pole as a crucial tool for tightrope walkers, it is also worth considering that, for example, circus performers are quite able to carry out shorter walks without the use of such tool. They typically use their arms or even their legs as a substitute: the arms are indeed stretched out sideways, thus increasing the moment of inertia in the coronal plane, with quick movements of the wrist/elbow of one arm, asymmetric with respect to the other arm, in order induce the small medio-lateral shifts of the body CoM; such motions are anti-correlated with respect to the rotations of the body axis and may be sufficient to maintain balance, if appropriately timed. This kind of behaviour may be observed also in the exercises on balance beam of artistic gymnastics, for example when the gymnasts attempt to recover balance after a somersault. Moreover, in this case, the gymnast may also use additional balancing gestures by moving sideways one leg or even the pelvis, in order to counteract the potential shift of the body CoM outside the small width of the balance beam. All such occurrences are examples of the CoM stabilization strategy, made necessary by the extremely limited range of variation of the CoP position. In general two different balancing concepts can be envisaged in the same COMS framework: a *static* balancing concept, namely balancing the gravity torque of the main part of the body with a counteracting gravity torque due to the shifted balancing pole or shifted body part; the alternative is the use of a *dynamic* balancing effect or counter-rotation mechanism. Such dynamic balancing behaviours can be observed in a number of situations: for example, tightrope walkers may perform slight rotations of the balancing pole in such a way to take advantage of the fact that the torque applied by the walker to the pole is reflected as an opposite torque on the body of the walker, according to the action-reaction law of physics; a counter-rotation mechanism has also been observed for the compensation of anteroposterior perturbations on the

control of the centre of mass during walking [40], and so on. In any case, the two balancing effects, static and dynamic, may be mixed in a synergistic way by an optimal tuning of the control parameters, in the context of a learning process, and this is an open field of research.

In the same line of thought, we may consider the study [16] that compared perturbed and unperturbed balance in handstand, in relation with standard upright standing: it was found that the dominant control strategy adopted by professional gymnasts in handstand is a wrist strategy, i.e. a special kind of *COPS* where the CoP is changed by activating the wrist muscles in such a way to anticipate the oscillations of the CoM, while keeping the body as straight as possible in such a way to emulate an inverted pendulum. However, several other strategies can also be employed to help maintain balance, particularly by non-professional performers. In particular, it is common to detect the emergence of elbow, shoulder or hip gestures (that indeed implement different *COMS* varieties): as balance in handstand becomes more precarious for lack of training, for disturbances or due to the fact that the physiological range of motion of the CoP in the sagittal plane is significantly reduced in handstand in comparison with upright stand, it becomes likely that these other control strategies may be employed, apparently violating the assumption of a single-segment inverted pendulum. In all the circumstances, however, the virtual CoP–CoM line drives the balancing gestures with one strategy or the other.

The two balancing strategies highlighted in this study as a general purpose computational machinery, emerging from the conquer of standard bipedal standing and then extended with some variation to a generality of more challenging balancing skills, resonate well with the general motor ability hypothesis [41]. Moreover, we may consider the study [42] that investigated the effect of expertise in gymnastics on generic postural control by comparing the size of postural sway in the standard bipedal and more challenging monopodal conditions, with or without vision, in elite gymnasts in comparison with non-gymnast controls: it was found that the difference between the two groups is null in bipedal standing, with or without vision, as well as in the more challenging monopodal condition, except in the non-vision case in which elite gymnasts perform much better. This can be explained by assuming that the effect of specific training in challenging balancing situations improves the capability to extract reliable information also from reduced sensory feedback but is unlikely to force the subjects to produce new control strategies. This consideration is also consistent with the fact that professional tightrope walkers can carry out blindfolded long challenging walks.

There are balancing skills where *COPS* simply cannot apply because the area of the support base tends to vanish. This is the case, for example, of classical ballet poses *en pointe* like *retiré*, *arabesque* or *penché* in which the BoS [43] is of the order of a few square centimetres. Professional ballerinas can keep such poses for several seconds (typically 3–5 s) without any external support or, apparently, compensatory arm movements. But is it for real or hiding it is a crucial aspect of the skill? The problem is that it is difficult to assess specific balance skills in dancers during performance of ballet movements, with the exception of tasks of limited difficulty like *balancé displacement*s [44] or *passé demi pointe* [45]. Therefore, only a few studies have discussed this subject in the framework of biomechanical analysis. In particular, a recent literature review [46] cites 89 papers from 1970 to 2009 related to biomechanical research in dance, mostly in classical ballet. Only one of them is devoted to arm movements [47]: by using a motion capture device the arm motions of professional ballerinas and amateur dancers were digitized while performing a sequence of the Swan Lake. Results indicated that in professional dancers the elbow moved first, and there was a wide range of movements at the elbow and wrist while the movements of amateur dancers had less variability, with little change in joint angles. Is this evidence of the organization of arm gestures in arabesque and similar balancing tasks en pointe by professional ballerinas for implementing the CoM strategy?

The proposed, preliminary simulation study has clear limitations primarily related to the simplifications of the model that were listed in the introduction, particularly in relation with the CoM stabilization strategy. Of course, relaxation or reduction of the simplifications might improve the biological plausibility of the model but also its complexity as well the difficulty to identify realistic values of the control parameters in the absence of a sufficiently large set of experimental data related to such balancing skills. The implementation of the CoM stabilization strategy used in the present simulation study is static: it aims at compensating the gravity torque of the body, referred to the (unmovable or scarcely movable) CoP, with a counteractive torque determined by a shifted balancing element, such as a pole or a limb (arm or leg). This mechanism is static because it ignores the dynamics of such balancing gestures. It was assumed that such simplification is compatible with the tightrope balancing paradigm where the presence of the massive balancing pole slows down drastically the dynamics of medio-lateral sway to the point of uncoupling the two stabilization processes (in the sagittal and coronal planes, respectively) and reducing the *COMS* to a static mechanism. However, it is quite possible that trained subjects can learn to integrate

the static with the dynamic stabilization strategy, related to some kind of counter-rotation mechanism [40], although integrating the two strategies would increase the complexity of the learning process due to the increased dimensionality of the control mechanism. The choice of very long balancing poles by professional tightrope walkers can indeed be interpreted as the attempt to slow down the medio-lateral oscillations to the point where the static stabilization strategy is sufficient to reliably avoid the danger of falling. In order to better understand the underlying motor control aspects, future research could follow two parallel lines: (i) the *experimental line*, namely collecting oscillatory data in a laboratory environment sufficiently similar to the tightrope situation with balancing tools of decreasing length in order to acquire empiric knowledge on the transition from the static to the dynamic strategy and on the modes of integration of the two control paradigms; and (ii) the *modelling line*, focusing on the modification of the intermittent control paradigm, proposed in this preliminary study, by adding a component related to a counter-rotation mechanism in order to match the experimental evidence.

Data accessibility. The Matlab simulation script that was used for generating the figures and for computing the numerical results discussed in the paper is available for download and immediate usage from the Dryad Digital Repository: https://doi.org/10.5061/dryad.q573n5tff [48]. Each simulation run lasts 240 s (changed by the user if required). For each run the random number generators are reset, thus the generated oscillatory patterns are independent. As a consequence, the reported graphs are just representative samples, slightly different from trial to trial, obtained by plotting q1 (body sway in the sagittal plane) and q2 (body sway in the coronal plane) and the corresponding derivatives.

Competing interests. I declare I have no competing interests.

Funding. I received no specific funding for this study.

Acknowledgements. This study was supported by RBCS (Robotics, Brain and Cognitive Science Department), Italian Institute of Technology, Genoa, Italy, in the framework of the general interest of the RBCS research unit for motor cognition in humans and robots.

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
