## [Reviewer comments · Royal Society Open Science]

Review History

RSOS-200111.R0 (Original submission)

Review form: Reviewer 1

Is the manuscript scientifically sound in its present form?

No

Are the interpretations and conclusions justified by the results?

No

Is the language acceptable?

Yes

Do you have any ethical concerns with this paper?

No

Have you any concerns about statistical analyses in this paper?

No

Recommendation?

Major revision is needed (please make suggestions in comments)

Comments to the Author(s)

Review: RSOS-200111

Title: CoP vs. CoM stabilization strategies: the tightrope balancing case

This manuscript describes a simulation study of a tightrope walker controlled in AP direction by an Ankle strategy and in ML direction by a CoM strategy. Overall this is an interesting study, and it is well written. However, I do have some concerns relating to the model used and the interpretation of the data, as outlined below.

Major

1. In equation 5, it seems that a linearized pendulum is assumed, because otherwise, both terms would have $\sin(q)$ instead of q . Why was a linearized pendulum used (why not a full pendulum?) and at the very least, it should be mentioned clearly in the text that this was done.
2. Why are reaction forces not taken into account? It seems that these may be quite important? Especially given that a displacement of the pole to the left would lead to a reaction force to the right, which would impose an opposite effect to the effect that is wanted?
3. Why is rotation of the beam not taken into account? It seems that in human balancing, the counter rotation mechanism (see Hof 2007) is thought of as important (see also one of our recent papers; van den Bogaart 2020)
4. The author used a simple euler integration, which, in general, is not suitable for integrating stochastic differential equations. I'm no expert on this, but know that for stochastic differential equations, more sophisticated integration procedures should be used?
5. Page 10 lines 9-10; here it is argued that the q_{ap} and q_{ml} are uncorrelated, which supports the hypotheses that these stabilization strategies can effectively function in an independent manner. However, it seems that this is quite logical here, as the Ap and ML equations are not coupled in any way, and hence, basically, 2 independent systems are simulated, both with a different controller. Hence, it should come at no surprise that the controllers function in an independent manner. Or do I oversee something here?
6. It seems that the last paragraph of the results is actually a more suitable paragraph for in the discussion?

Minor

1. It's odd that the authors don't cite Hof et al (2007), which seems to fit perfectly in the first paragraph of the introduction
2. Page 3, line 47: "In front" □ "in front"
3. Page 4 line 33 (and maybe other places) : "we" □ "I" as it's a single author paper
4. Page 7 line 40; it's unclear what the q_p means, also not explained in the text?
5. Page 7, equation 10; what lowpass filter was used? What cut off etc? I realize that this is mentioned in the table with parameters, but it may be good to also mention it here.
6. Page 7 line 49; it would be good to have a plot of the phase plane, with indications of stable and unstable regions, and some indication of how the controller acts.
7. Page 8, line 10; the feedback delay is set at 200ms, yet there are short term reflexes that could be quicker, would this change the results much?
8. Page 9, line 30 or so: It would be good to show for both controllers the imposed torques, so that the reader gets a better idea about what actually is happening?
9. Figure 5 AP and ML are switched
10. I was able to run the supplied code without any problems, but it did not directly lead to plots of the results, and it may also not be immediately obvious how to do so. It would be great if the code could be extended, such that such plots are also generated.

References

van den Bogaart M, Bruijn SM, van Dieën JH, Meyns P. The effect of anteroposterior perturbations on the control of the center of mass during treadmill walking. *J Biomech.* 2020;103:109660. doi:10.1016/j.jbiomech.2020.109660

Hof AL. The equations of motion for a standing human reveal three mechanisms for balance. J Biomech. 2007;40(2):451-457. doi:10.1016/j.jbiomech.2005.12.016

Review form: Reviewer 2

Is the manuscript scientifically sound in its present form?

Yes

Are the interpretations and conclusions justified by the results?

No

Is the language acceptable?

Yes

Do you have any ethical concerns with this paper?

No

Have you any concerns about statistical analyses in this paper?

No

Recommendation?

Accept with minor revision (please list in comments)

Comments to the Author(s)

Please see attached file (Appendix A).

Decision letter (RSOS-200111.R0)

Dear Dr Morasso,

The editors assigned to your paper ("CoP vs. CoM stabilization strategies: the tightrope balancing case") have now received comments from reviewers. We would like you to revise your paper in accordance with the referee and Associate Editor suggestions which can be found below (not including confidential reports to the Editor). Please note this decision does not guarantee eventual acceptance.

Please submit a copy of your revised paper before 26-Jul-2020. Please note that the revision deadline will expire at 00.00am on this date. If we do not hear from you within this time then it will be assumed that the paper has been withdrawn. In exceptional circumstances, extensions may be possible if agreed with the Editorial Office in advance. We do not allow multiple rounds of revision so we urge you to make every effort to fully address all of the comments at this stage. If deemed necessary by the Editors, your manuscript will be sent back to one or more of the original reviewers for assessment. If the original reviewers are not available, we may invite new reviewers.

- Data accessibility

If you wish to submit your supporting data or code to Dryad (<http://datadryad.org/>), or modify your current submission to dryad, please use the following link:
<http://datadryad.org/submit?journalID=RSOS&manu=RSOS-200111>

- Competing interests

- Authors' contributions

- Acknowledgements

- Funding statement

on behalf of R. Kerry Rowe (Subject Editor)
openscience@royalsociety.org

Associate Editor's comments:

Comments to the Author:

Thank you for your patience while we sought reviewers at this difficult time - as we're sure you can imagine, many reviewers have faced extreme pressures on their time (moving to online teaching, remote working, home schooling or caring responsibilities, etc.), and the journal is grateful for the support of the referees who have provided substantial reviews here.

Based on the reviewers' commentary, we would like you to revise the manuscript, and the revision and associated point-by-point reply will be sent back to at least one, more likely both, of the reviewers for their view of the changes made. With this in mind, we urge you to take care to be sure you have fully responded to the referees' concerns before resubmitting.

Reviewers' Comments to Author:

Reviewer: 1

Comments to the Author(s)

Review: RSOS-200111

Title: CoP vs. CoM stabilization strategies: the tightrope balancing case

This manuscript describes a simulation study of a tightrope walker controlled in AP direction by an Ankle strategy and in ML direction by a CoM strategy. Overall this is an interesting study, and it is well written. However, I do have some concerns relating to the model used and the interpretation of the data, as outlined below.

Major

1. In equation 5, it seems that a linearized pendulum is assumed, because otherwise, both terms would have $\sin(q)$ instead of q . Why was a linearized pendulum used (why not a full pendulum?) and at the very least, it should be mentioned clearly in the text that this was done.
2. Why are reaction forces not taken into account? It seems that these may be quite important? Especially given that a displacement of the pole to the left would lead to a reaction force to the right, which would impose an opposite effect to the effect that is wanted?

3. Why is rotation of the beam not taken into account? It seems that in human balancing, the counter rotation mechanism (see Hof 2007) is thought of as important (see also one of our recent papers; van den Bogaart 2020)
4. The author used a simple euler integration, which, in general, is not suitable for integrating stochastic differential equations. I'm no expert on this, but know that for stochastic differential equations, more sophisticated integration procedures should be used?
5. Page 10 lines 9-10; here it is argued that the q_{ap} and q_{ml} are uncorrelated, which supports the hypotheses that these stabilization strategies can effectively function in an independent manner. However, it seems that this is quite logical here, as the AP and ML equations are not coupled in any way, and hence, basically, 2 independent systems are simulated, both with a different controller. Hence, it should come at no surprise that the controllers function in an independent manner. Or do I oversee something here?
6. It seems that the last paragraph of the results is actually a more suitable paragraph for in the discussion?

Minor

1. It's odd that the authors don't cite Hof et al (2007), which seems to fit perfectly in the first paragraph of the introduction
2. Page 3, line 47: "In front" \square "in front"
3. Page 4 line 33 (and maybe other places) : "we" \square "I" as it's a single author paper
4. Page 7 line 40; it's unclear what the q_p means, also not explained in the text?
5. Page 7, equation 10; what lowpass filter was used? What cut off etc? I realize that this is mentioned in the table with parameters, but it may be good to also mention it here.
6. Page 7 line 49; it would be good to have a plot of the phase plane, with indications of stable and unstable regions, and some indication of how the controller acts.
7. Page 8, line 10; the feedback delay is set at 200ms, yet there are short term reflexes that could be quicker, would this change the results much?
8. Page 9, line 30 or so: It would be good to show for both controllers the imposed torques, so that the reader gets a better idea about what actually is happening?
9. Figure 5 AP and ML are switched
10. I was able to run the supplied code without any problems, but it did not directly lead to plots of the results, and it may also not be immediately obvious how to do so. It would be great if the code could be extended, such that such plots are also generated.

References

van den Bogaart M, Bruijn SM, van Dieën JH, Meyns P. The effect of anteroposterior perturbations on the control of the center of mass during treadmill walking. *J Biomech.* 2020;103:109660. doi:10.1016/j.jbiomech.2020.109660

Hof AL. The equations of motion for a standing human reveal three mechanisms for balance. *J Biomech.* 2007;40(2):451-457. doi:10.1016/j.jbiomech.2005.12.016

Reviewer: 2

Comments to the Author(s)

Please see attached file. (Review for RSOS-200111.pdf)

Author's Response to Decision Letter for (RSOS-200111.R0)

See Appendix B.

RSOS-200111.R1 (Revision)

Review form: Reviewer 1

Is the manuscript scientifically sound in its present form?

Yes

Are the interpretations and conclusions justified by the results?

Yes

Is the language acceptable?

Yes

Do you have any ethical concerns with this paper?

Yes

Have you any concerns about statistical analyses in this paper?

No

Recommendation?

Accept with minor revision (please list in comments)

Comments to the Author(s)

Previous comment 8. Page 9, line 30 or so: It would be good to show for both controllers the imposed torques, so that the reader gets a better idea about what actually is happening?

Author answer: As a matter of fact the control torques are plotted in figure 4 (now figure 5). Panel B compares, for the ML controller, the control torque, with the destabilizing gravity torque. Panel A shows, for the AP controller, the same information because the CoP position is proportional to the control torque and the CoM position to the gravity torque.

New comment: but why not show the torques for panel a? This way, the figure is more similar to figure b.

Previous comment 9. Figure 5 AP and ML are switched.

Author answer: I don't think so.

New comment; my meaning was that in all previous figures, AP was plotted first, and then ML. In figure 5 (now 6), AP is on the right, and ML on the left. This may be confusing, as AP is always on the left.

Previous comment 10. I was able to run the supplied code without any problems, but it did not directly lead to plots of the results, and it may also not be immediately obvious how to do so. It would be great if the code could be extended, such that such plots are also generated.

Author answer: I must say that there is no way to extend the code in order to reproduce exactly the plotted patterns. With reference to the answer to the major question no. 4, the numerical integration of the equations of sway in the two planes is based on repeated random sampling to obtain numerical results. Thus any simulation run provides independent examples of the sway patterns. The patterns illustrated in the figures are just a representative example of the underlying stochastic process. I added a sentence about that at the beginning of the Results section.

New comment: but if the plots were generated at the end of the script, this would already give the interested reader more idea of which variables are what, and how plots are generated. In addition, by setting the random number generator to a certain state, it should be possible to create a script that exactly produces the figures as they are seen in the paper.

Review form: Reviewer 2

Is the manuscript scientifically sound in its present form?

Yes

Are the interpretations and conclusions justified by the results?

Yes

Is the language acceptable?

Yes

Do you have any ethical concerns with this paper?

No

Have you any concerns about statistical analyses in this paper?

No

Recommendation?

Accept with minor revision (please list in comments)

Comments to the Author(s)

Two minor comments about Figure 3:

(1) units on the y axes - should they be deg/sec?

(2) author should consider using same range along x and y axes for the two subplots - it might help in comparing the two plots.

All other concerns have been addressed.

I am excited by the ideas in this paper, and I recommend that the manuscript be accepted for publication.

Decision letter (RSOS-200111.R1)

Dear Dr Morasso

On behalf of the Editors, we are pleased to inform you that your Manuscript RSOS-200111.R1 "CoP vs. CoM stabilization strategies: the tightrope balancing case" has been accepted for publication in Royal Society Open Science subject to minor revision in accordance with the referees' reports. Please find the referees' comments along with any feedback from the Editors below my signature.

Please submit your revised manuscript and required files (see below) no later than 7 days from today's (ie 10-Aug-2020) date. Note: the ScholarOne system will 'lock' if submission of the

revision is attempted 7 or more days after the deadline. If you do not think you will be able to meet this deadline please contact the editorial office immediately.

on behalf of Prof R. Kerry Rowe (Subject Editor)
openscience@royalsociety.org

Associate Editor Comments to Author:

Thank you for so constructively engaging with the reviewers' and the editor's comments. A handful of changes are left to be incorporated before the paper may be accepted.

Reviewer comments to Author:

Reviewer: 2

Comments to the Author(s)

Two minor comments about Figure 3:

(1) units on the y axes - should they be deg/sec?

(2) author should consider using same range along x and y axes for the two subplots - it might help in comparing the two plots.

All other concerns have been addressed.

I am excited by the ideas in this paper, and I recommend that the manuscript be accepted for publication.

Reviewer: 1

Comments to the Author(s)

Previous comment 8. Page 9, line 30 or so: It would be good to show for both controllers the imposed torques, so that the reader gets a better idea about what actually is happening?

Author answer: As a matter of fact the control torques are plotted in figure 4 (now figure 5). Panel B compares, for the ML controller, the control torque, with the destabilizing gravity torque. Panel A shows, for the AP controller, the same information because the CoP position is proportional to the control torque and the CoM position to the gravity torque.

New comment: but why not show the torques for panel a? This way, the figure is more similar to figure b.

Previous comment 9. Figure 5 AP and ML are switched.

Author answer: I don't think so.

New comment; my meaning was that in all previous figures, AP was plotted first, and then ML. In figure 5 (now 6), AP is on the right, and ML on the left. This may be confusing, as AP is always on the left.

Previous comment 10. I was able to run the supplied code without any problems, but it did not directly lead to plots of the results, and it may also not be immediately obvious how to do so. It would be great if the code could be extended, such that such plots are also generated.

Author answer: I must say that there is no way to extend the code in order to reproduce exactly the plotted patterns. With reference to the answer to the major question no. 4, the numerical integration of the equations of sway in the two planes is based on repeated random sampling to obtain numerical results. Thus any simulation run provides independent examples of the sway patterns. The patterns illustrated in the figures are just a representative example of the underlying stochastic process. I added a sentence about that at the beginning of the Results section.

New comment: but if the plots were generated at the end of the script, this would already give the interested reader more idea of which variables are what, and how plots are generated. In addition, by setting the random number generator to a certain state, it should be possible to create a script that exactly produces the figures as they are seen in the paper.

===PREPARING YOUR MANUSCRIPT===

===PREPARING YOUR REVISION IN SCHOLARONE===

Author's Response to Decision Letter for (RSOS-200111.R1)

See Appendix C.

Decision letter (RSOS-200111.R2)

Dear Dr Morasso,

It is a pleasure to accept your manuscript entitled "CoP vs. CoM stabilization strategies: the tightrope balancing case" in its current form for publication in Royal Society Open Science.

on behalf of the Associate Editor and Professor R. Kerry Rowe (Subject Editor)
openscience@royalsociety.org

Appendix A

Review for RSOS-200111.

The author has outlined an interesting mechanism for upright balance control for the case when the balance task constrains the COP under the feet to have minimal motion in certain directions (ML in the present manuscript). The key idea is that when the COP cannot function as a control variable due to the above constraint, the COM is controlled instead via the movements of the body segments. The example used in the manuscript is tightrope walking, and the COM is controlled by moving the pole that the walker carries. The author provides simulation data only, and speculates about the application of this notion to human movements.

This is an interesting proposal. Barring a few points of confusion regarding interpretation and extension of this idea, the application of the proposal to a variety of movements that may be considered as 'non-traditional' to biomechanics and motor control communities makes this manuscript quite attractive for publication.

The points of confusion that are pointed out first. This is followed by requests for minor clarifications.

Overall, this reviewer found the second and the fourth paragraphs of the Discussion a bit discursive, and not very helpful.

The first and main point of confusion is the claim that the COPS vs COMS strategies are a generalization of the ankle vs hip strategy for sway control in the sagittal plane. The author makes this claim in the Discussion (page 11, and again on page 12 in the context of hand stands). This generalization is under-developed, and frankly, does not seem necessary for this manuscript. First, on what basis is hip engagement for maintaining balance considered a COMS strategy? Does engaging hip motion/torques (presumably in addition to ankle involvement) change COM only? Generating hip torques could change COP through inter-joint coupling. Second, the author switches from the decoupling of planes of motion (frontal vs sagittal) in the presented model to two strategies elucidated in the sagittal plane. It seems that this claim will need evidence, and such evidence is not provided in this manuscript.

The second point of confusion is that the author states: 'The two angular oscillations (q_{ap} and q_{ml}) are uncorrelated, supporting the hypothesis that the two stabilization strategies can effectively function in an independent manner.' This is confusing because the model decouples the movements in the two planes. Isn't it obvious that the output coordinates would be uncorrelated? Furthermore, it is known that delay-differential systems can be stabilized with feedback. So, using this data (or any simulation data presented in this manuscript) to support this 'hypothesis' is neither interesting nor illuminating.

Minor clarifications:

1. Please explain the dis/activation condition in equations 9 and 10.
2. This reviewer was unable to follow the 'Control Action' in equation 10. Please explain.
3. This reviewer was also unable to understand 'alpha': the lines 49-55 on page 7 are confusing.
4. Page 10-11: "We found that with a 60% pole reduction it was impossible to avoid a quick fall and for a 70% reduction stability could be achieved but only for limited stretches of time (30-60 s)." How come a larger reduction in pole length resulted in stable behavior? Is this a typo?

Appendix B

Anita Kristiansen

Editorial Coordinator - Royal Society Open Science

Review: RSOS-200111

Title: CoP vs. CoM stabilization strategies: the tightrope balancing case

Associate Editor's comments:

Comments to the Author:

Thank you for your patience while we sought reviewers at this difficult time - as we're sure you can imagine, many reviewers have faced extreme pressures on their time (moving to online teaching, remote working, home schooling or caring responsibilities, etc.), and the journal is grateful for the support of the referees who have provided substantial reviews here.

Based on the reviewers' commentary, we would like you to revise the manuscript, and the revision and associated point-by-point reply will be sent back to at least one, more likely both, of the reviewers for their view of the changes made. With this in mind, we urge you to take care to be sure you have fully responded to the referees' concerns before resubmitting.

I read carefully the reviewers' commentary, all of them well formulated and useful for improving the manuscript. I did my best to respond to modify the manuscript which is included in the revised version. The point by point answers (highlighted in yellow) to the reviewers' concerns are included.

Reviewer #1

Comments to the Author(s)

Review: RSOS-200111

Title: CoP vs. CoM stabilization strategies: the tightrope balancing case

This manuscript describes a simulation study of a tightrope walker controlled in AP direction by an Ankle strategy and in ML direction by a CoM strategy. Overall this is an interesting study, and it is well written. However, I do have some concerns relating to the model used and the interpretation of the data, as outlined below.

I am quite grateful to this reviewer for his/her deep insight and useful suggestions.

Major

1. In equation 5, it seems that a linearized pendulum is assumed, because otherwise, both terms would have $\sin(q)$ instead of q . Why was a linearized pendulum used (why not a full pendulum?) and at the very least, it should be mentioned clearly in the text that this was done.

Yes, the pendulum equation is linearized for simplicity. This is mentioned in the text. Considering that the simulations shows that both angles are consistently smaller than 1 deg changing q with $\sin(q)$ would not make any difference.

2. Why are reaction forces not taken into account? It seems that these may be quite important? Especially given that a displacement of the pole to the left would lead to a reaction force to the right, which would impose an opposite effect to the effect that is wanted?

As stated in the introduction the model includes several simplifications, focusing on the feasibility of the intermittent control hypothesis for the stabilization in the coronal plane. In the preliminary version of the manuscript I listed 4 simplifications. In the review I add two. The fifth is related to the reaction forces that are neglected for two reasons, derived from the observation of videos of famous long distance tightrope walkers like Nick Wallenda: 1) the lateral movements of the pole are very slow, 2) as shown in the picture below, funambulists typically unload the weight of the pole with a strap on the shoulder and this attachment might have a damping effect on the reaction force. However, in principle the reaction force effect can be introduced in the model for a future deeper analysis.

3. Why is rotation of the beam not taken into account? It seems that in human balancing, the counter rotation mechanism (see Hof 2007) is thought of as important (see also one of our recent papers; van den Bogaart 2020).

The answer is similar to the previous one. This is the sixth simplification of the model, motivated by the fact that observing videos of long distance walkers it appears that, for example in the case of Nick Wallenda walking over the Niagara falls, the pole is barely rotated during the more than 10 minutes walk. By increasing as much as possible the falling time constant (with a very long pole) it appears that the funambulist can also maximize the slowing down of the control action, reducing it to the quasi-static component. Of course there are balancing tasks, like slackline balancing, where the dynamics of the balancing action is strongly increased and in that case reaction forces or torques may be predominant.

4. The author used a simple Euler integration, which, in general, is not suitable for integrating stochastic differential equations. I'm no expert on this, but know that for stochastic differential equations, more sophisticated integration procedures should be used?

The problem is not the Euler method per se, which is the simplest method of numerical integration of differential equations. Runge-Kutta or predictor-corrector methods would face the same criticism, due to the fact that, in principle, the solution of equation 4, i.e. $q(t)$, is a stochastic process. The sophisticated techniques of integration of stochastic differential equation (SDE) are difficult to apply in our case due to the strong non linearity introduced by the intermittent controller. The solution adopted in the simulation experiments is based on the rationale of Monte Carlo methods, i.e. computational algorithms that rely on repeated random sampling to obtain numerical results which allow to estimate relevant statistic indicators of the underlying stochastic process.

5. Page 10 lines 9-10; here it is argued that the q_{ap} and q_{ml} are uncorrelated, which supports the hypotheses that these stabilization strategies can effectively function in an independent manner. However, it seems that this is quite logical here, as the A_p and M_L equations are not coupled in any way, and hence, basically, 2 independent systems are simulated, both with a different controller. Hence, it should come at no surprise that the controllers function in an independent manner. Or do I oversee something here?

As a matter of fact there is a direct, but small coupling effect due to the fact that the inertia matrix is not diagonal, although I admit that the off-diagonal terms are much smaller than the diagonal ones. However, the intermittency of the control actions might induce, in principle, some kind of dangerous entrainment of the two oscillatory patterns that apparently does not occur in the simulation experiments.

6. It seems that the last paragraph of the results is actually a more suitable paragraph for in the discussion?

Thank you for the suggestion. I broke this paragraph in two parts. The first one is left in the results and the second one is shifted to the re-written discussion.

Minor

1. It's odd that the authors don't cite Hof et al (2007), which seems to fit perfectly in the first paragraph of the introduction.

Good suggestion, thank you.

2. Page 3, line 47: "In front" - "in front"

Corrected, thank you.

3. Page 4 line 33 (and maybe other places) : "we" or "I" as it's a single author paper.

Thank you for the observation. Actually I intended to use the royal we but this is not appropriate. However, using the first person implies an expression of subjective will, which perhaps is not appropriate either. Thus I turned to the passive form in the third person when I referred to a choice, as in the quoted example, and I kept we when referring to a fact or an opinion that may be shared by the community of researchers.

4. Page 7 line 40; it's unclear what the q_p means, also not explained in the text?

As a matter of fact q_p is shown in figure 1. However, as suggested, I added the definition in the text: q_p is the angle between the line from the rope to the CoM of the pole and the vertical.

5. Page 7, equation 10; what lowpass filter was used? What cut off etc? I realize that this is mentioned in the table with parameters, but it may be good to also mention it here.

The transfer functions of the filter is $F(s) = \frac{1}{(s/\omega_f)^2 + 2\xi(s/\omega_f) + 1}$

6. Page 7 line 49; it would be good to have a plot of the phase plane, with indications of stable and unstable regions, and some indication of how the controller acts.

Good suggestion. Instead of adding a new figure I replotted the old figure 2 (now figure 3) color coding the segments related to the off-phases and on-phases respectively. Please note the slight difference between the shapes of the old figure 2 and the new figure 3, due to the random sampling issue, named in my answer to your question 10.

7. Page 8, line 10; the feedback delay is set at 200ms, yet there are short term reflexes that could be quicker, would this change the results much?

This is hard to say. Certainly there are quick reflexes that do not involve cortical processing and are probably associated to limited/local groups of muscles, for example for maintaining the approximate alignment of the body inverted pendulum. The postulated intermittent controller operates at a higher, global level and a feedback delay of the order of 200 ms is reasonable. In any case, one of the purposes of the simulation study is to show that even with such a large delay the system can be stabilized with a slow bounded stability.

8. Page 9, line 30 or so: It would be good to show for both controllers the imposed torques, so that the reader gets a better idea about what actually is happening?

As a matter of fact the control torques are plotted in figure 4 (now figure 5). Panel B compares, for the ML controller, the control torque, with the destabilizing gravity torque. Panel A shows, for the AP controller, the same information because the CoP position is proportional to the control torque and the CoM position to the gravity torque.

9. Figure 5 AP and ML are switched.

I don't think so.

10. I was able to run the supplied code without any problems, but it did not directly lead to plots of the results, and it may also not be immediately obvious how to do so. It would be great if the code could be extended, such that such plots are also generated.

I must say that there is no way to extend the code in order to reproduce exactly the plotted patterns. With reference to the answer to the major question no. 4, the numerical integration of the equations of sway in the two planes is based on repeated random sampling to obtain numerical results. Thus any simulation run provides independent examples of the sway patterns. The patterns illustrated in the figures are just a representative example of the underlying stochastic process. I added a sentence about that at the beginning of the Results section.

Reviewer #2

Comments to the Author(s)

Review: RSOS-200111

Title: CoP vs. CoM stabilization strategies: the tightrope balancing case. The author has outlined an interesting mechanism for upright balance control for the case when the balance task constrains the COP under the feet to have minimal motion in certain directions (ML in the present manuscript). The key idea is that when the COP cannot function as a control variable due to the above constraint, the COM is controlled instead via the movements of the body segments. The example used in the manuscript is tightrope walking, and the COM is controlled by moving the pole that the walker carries. The author provides simulation data only, and speculates about the application of this notion to human movements.

This is an interesting proposal. Barring a few points of confusion regarding interpretation and extension of this idea, the application of the proposal to a variety of movements that may be considered as 'nontraditional' to biomechanics and motor control communities makes this manuscript quite attractive for publication.

I thank this reviewer for identifying the weak points of my work and helping me to improve it in a significant manner.

The points of confusion that are pointed out first. This is followed by requests for minor clarifications.

Overall, this reviewer found the second and the fourth paragraphs of the Discussion a bit discursive, and not very helpful.

The Discussion session has been re-written in order to take into account the criticism and suggestions of both reviewers.

The first and main point of confusion is the claim that the COPS vs COMS strategies are a generalization of the ankle vs hip strategy for sway control in the sagittal plane. The author makes this claim in the Discussion (page 11, and again on page 12 in the context of hand stands). This generalization is underdeveloped, and frankly, does not seem necessary for this manuscript. First, on what basis is hip engagement for maintaining balance considered a COMS strategy? Does engaging hip motion/torques (presumably in addition to ankle involvement) change COM only? Generating hip torques could change COP through interjoint coupling. Second, the author switches from the decoupling of planes of motion (frontal vs sagittal) in the presented model to two strategies elucidated in the sagittal plane. It seems that this claim will need evidence, and such evidence is not provided in this manuscript.

Hip engagement for maintaining balance is an example of utilizing the COMS strategy when environmental conditions (e.g. limited size of the support basis, soft standing surface) or physiological conditions (e.g. in the case of elderly people) reduce significantly the feasible range of motion of the CoP. However, this does not imply that, in general, engaging hip motion/torques mostly changes the CoM: hip torques could indeed change CoP through interjoint coupling if the BoS is sufficiently large. In other words, the clearcut distinction between COPS and COMS is that in the former case the oscillation of the CoP is larger and anticipates the oscillation of the CoM, whereas in the latter case it is the other way around and it achieves stability by engaging different body parts.

The second point of confusion is that the author states: ‘**The two angular oscillations (q_{ap} and q_{ml}) are uncorrelated, supporting the hypothesis that the two stabilization strategies can effectively function in an independent manner.**’ This is confusing because the model decouples the movements in the two planes. Isn’t it obvious that the output coordinates would be uncorrelated? Furthermore, it is known that delay-differential systems can be stabilized with feedback. So, using this data (or any simulation data presented in this manuscript) to support this ‘hypothesis’ is neither interesting nor illuminating.

As a matter of fact there is a direct, but small coupling effect due to the fact that the inertia matrix is not diagonal, although I admit that the off-diagonal terms are much smaller than the diagonal ones. However, the intermittency of the control actions might induce, in principle, some kind of dangerous entrainment of the two oscillatory patterns that apparently does not occur in the simulation experiments.

Minor clarifications:

1. Please explain the dis/activation condition in equations 9 and 10.

The activation/dis-activation conditions of equations 9 and 10 correspond to the fact that the trajectory of the delayed state vector enters one region coming from the other (see figure 2): from safe to unsafe or vice versa. Please refer to the new figure 2, specifically added for clarifying the point.

2. This reviewer was unable to follow the ‘Control Action’ in equation 10. Please explain.

The following text is inserted after equation 10.

The control variable $\gamma(t)$ consists of displacing the pole CoM in the opposite direction of the body CoM disequilibrium ($\gamma = -(Pq + D\dot{q})$) during the on-phase, whereas the two CoMs are kept aligned in the off-phase ($\gamma = 0$). Since this is a discontinuous variable it will be smoothed out by the arm control system. For simplicity, this smoothing is implemented in the simulation package by a simple low-pass filter (LPF), which is characterized by the following transfer function: $F(s) = \frac{1}{(s/\omega_f)^2 + 2\xi(s/\omega_f) + 1}$. The control action, namely the generation of the torque T_{ml} supposed to compensate the destabilizing effect of gravity, is then the biomechanical consequence of this controlled shift.

3. This reviewer was also unable to understand ‘alpha’: the lines 49-55 on page 7 are confusing.

The rationale of alpha is explained in the new figure 2.

4. Page 10-11: “**We found that with a 60% pole reduction it was impossible to avoid a quick fall and for a 70% reduction stability could be achieved but only for limited stretches of time (30-60 s).**” How come a larger reduction in pole length resulted in stable behavior? Is this a typo?

It is not a typo but a badly expressed evaluation: by “70% reduction” I intended “reduction of the pole length to 70% of the original length”

Appendix C

Reviewer: 1

Thank you for your patience and attention

Comments to the Author(s)

Previous comment 8. Page 9, line 30 or so: It would be good to show for both controllers the imposed torques, so that the reader gets a better idea about what actually is happening?

Author answer: As a matter of fact the control torques are plotted in figure 4 (now figure 5). Panel B compares, for the ML controller, the control torque, with the destabilizing gravity torque. Panel A shows, for the AP controller, the same information because the CoP position is proportional to the control torque and the CoM position to the gravity torque.

New comment: but why not show the torques for panel a? This way, the figure is more similar to figure b.

Good suggestion: I added a second scale on panel A that allows the comparison among the two oscillations in terms of torque.

Previous comment 9. Figure 5 AP and ML are switched.

Author answer: I don't think so.

New comment; my meaning was that in all previous figures, AP was plotted first, and then ML. In figure 5 (now 6), AP is on the right, and ML on the left. This may be confusing, as AP is always on the left.

Agreed. Thank you for the suggestion. My previous answer was due to a misunderstanding of your comment.

Previous comment 10. I was able to run the supplied code without any problems, but it did not directly lead to plots of the results, and it may also not be immediately obvious how to do so. It would be great if the code could be extended, such that such plots are also generated.

Author answer: I must say that there is no way to extend the code in order to reproduce exactly the plotted patterns. With reference to the answer to the major question no. 4, the numerical integration of the equations of sway in the two planes is based on repeated random sampling to obtain numerical results. Thus any simulation run provides independent examples of the sway patterns. The patterns illustrated in the figures are just a representative example of the underlying stochastic process. I added a sentence about that at the beginning of the Results section.

New comment: but if the plots were generated at the end of the script, this would already give the interested reader more idea of which variables are what, and how plots are generated. In addition, by setting the random number generator to a certain state, it should be possible to create a script that exactly produces the figures as they are seen in the paper.

I added the following sentence in the Data Availability section: "Each simulation run lasts 240 s (changed by the user if required). For each run the random number generators are reset, thus the generated oscillatory patterns are independent. As a consequence, the reported graphs are just representative

samples, slightly different from trial to trial, obtained by plotting q_1 (body sway in the sagittal plane) and q_2 (body sway in the coronal plane) and the corresponding derivatives.”

Reviewer comments to Author:

Reviewer: 2

Comments to the Author(s)

Two minor comments about Figure 3:

(1) units on the y axes - should they be deg/sec?

Correct, thank you for the indication.

(2) author should consider using same range along x and y axes for the two subplots - it might help in comparing the two plots.

Your suggestion is a reasonable possibility, but I preferred to leave the two subplots with different scales for purely graphical reasons because the B subplot would have been strongly compressed. For taking into account your concern I added a note in the figure caption. Moreover, figure 4 captures at least one part of your suggestion because the two oscillations are plotted on purpose with the same scale in order to enhance the different ranges.

All other concerns have been addressed.

Thanks for your patience and attention.

I am excited by the ideas in this paper, and I recommend that the manuscript be accepted for publication.